# A Study of a Gain Based Approach for Query Aspects in Recall Oriented Tasks

Giorgio Maria Di Nunzio *,† and Guglielmo Faggioli †

Department of Information Engineering, University of Padova, Via Gradenigo 6/b, 35131 Padova, Italy;
guglielmo.faggioli@phd.unipd.it
* Correspondence: giorgiomaria.dinunzio@unipd.it
† Authors contributed equally to this work.

**Abstract:** Evidence-based healthcare integrates the best research evidence with clinical expertise in order to make decisions based on the best practices available. In this context, the task of collecting all the relevant information, a recall oriented task, in order to take the right decision within a reasonable time frame has become an important issue. In this paper, we investigate the problem of building effective Consumer Health Search (CHS) systems that use query variations to achieve high recall and fulfill the information needs of health consumers. In particular, we study an intent-aware gain metric used to estimate the amount of missing information and make a prediction about the achievable recall for each query reformulation during a search session. We evaluate and propose alternative formulations of this metric using standard test collections of the CLEF 2018 eHealth Evaluation Lab CHS.

**Keywords:** query variations; query reformulations; query performance prediction; systematic reviews





## 1. Introduction

The study of the query representation in Information Retrieval has driven a lot of interest in recent years [1–7]. Several works in the past [8–10] showed the positive effect on the retrieval results of fusing runs retrieved with human-made multiple formulations of the same information need. Recent studies have shown how query reformulations automatically extracted from query logs can be as effective as those manually created by users [11]. Furthermore, the performance of a system can greatly improve when the "right" formulation of an information need is selected [4,5]. One of the main challenges in this research area is being able to suggest the best performing query (or queries) among the possible variations [4,5,12–14]. For example, Thomas et al. [4] observed that, the most prominent effect in predicting the performance of a query formulation is due to the information need and not to the "query wording". In this sense, query performance predictors actually predict the complexity of the information need, rather than the one the query itself. Zendel et al. [5] pursue a slightly different task. Following the literature on reference lists [15,16] they try to predict the performance for a query using information about queries representing the same information need. Benham et al. [3] define a fusion approach for multiple query formulations based on the concept of "topic centroid", which describes the information need as combination of its formulations. Dang et al. [12] address also the problem of improving the ranking results through a query formulation selection phase. Note that, Dang et al. [12] show how they are often capable of putting the best query in the first two positions (not only the first one), a further evidence of the complexity of the task.

A use case of query performance prediction is the systematic compilation of literature review. In fact, systematic reviews are scientific investigations that use strategies to include a comprehensive search of all potentially relevant articles. As time and resources are limited for compiling a systematic review, limits to the search are needed: for example, one may want to estimate how far the horizon of the search should be (i.e., all possible

cases/documents that could exist in the literature) in order to stop before the resources are finished [17]. Scells et al. [13] apply several state-of-the-art Query Performance Predictors to select the best query in the Systematic Reviews domain. They show how current Query Performance Prediction approaches perform poorly on this specific task. International evaluation campaigns have organized labs in order to study this problem in terms of the evaluation, through controlled simulation, of methods designed to achieve very high recall [18,19]. The CLEF initiative (http://www.clef-initiative.eu, accessed on 15 February 2021) has promoted the eHealth track since 2013 and, the CLEF 2018 eHealth Evaluation Lab Consumer Health Search (CHS) task [20] investigated the problem of building search engines that are robust to query variations to support information needs of health consumers.

In this paper, we study an alternative formulation of the intent-aware metric proposed by Umemoto et al. [21], in which the authors analyze a metric to estimate the amount of missing information for each query reformulation during a search session. Note that in [21] the authors do not propose an approach capable of predicting the recall of different formulations. Nevertheless, our perception is that, their approach can be easily adapted with good results also to the predictive task. In our case, our research goal is to understand whether a gain based approach can be used to predict the relative importance of each reformulation in terms of recall performance, in the context of Consumer Health Search where users need support for medical information needs.

In this sense, with respect to [21], our contribution is two-fold:

- we show that it is possible to apply the GAIN measure proposed in [21] to obtain a recall predictor over a set of formulations for the same topic;
- we furthermore show how to improve the results of such predictor by exploiting also the information obtained through the various formulations.

The paper is organized as follows: in Section 2, we present the original gain metric, while in Section 3 we define our alternative version to predict the performance in a recall-oriented fashion. In Section 4, we discuss the experimental analysis and results; while in Section 5 we give our final remarks.

## 2. A GAIN-Based Approach

In Umemoto et al. [21], define the intent-aware gain metric and the requirements that it should satisfy. They identify the following properties: importance, documents relevant to a central aspect of the search topic produce higher gain than those relevant to a peripheral one; relevance, highly relevant documents produce higher gain than partially relevant ones; novelty, documents relevant to an unexplored aspect produce higher gain than those relevant to a fully explored aspect.

The set of aspects $A_t$ of a topic $t$ is estimated through the process described in [22]: first, a set of subtopics $S_t$ is mined given a topic $t$; then, the subtopics are grouped into a set of clusters $C_t$. These clusters are regarded as the "facets" (We use *facets* instead of *aspects* to not repeat the same term that will be use to identify the most representative subtopic.) of $t$. The most representative subtopic $s$ is chosen from each cluster as formulation of the topic aspect $a$ using the formula $a = argmax_{s \in C_t} \mathrm{Imp}_t(s)$, where the importance of a subtopic $s$ is defined as:

$$\mathrm{Imp}_t(s) = \sum_{d \in D_s^N \cap d \in D_t^N} \frac{1}{\mathrm{Rank}_t(d)} \tag{1}$$

$D_s^N$ and $D_t^N$ denote the sets of the top $N$ retrieved documents for a subtopic $s$ and the topic $t$, respectively, and $\mathrm{Rank}_t(d)$ is the rank of the document $d$ in the ranked list for $t$.

It is crucial to stress that the definition of *importance*, and the following definition of *gain*, derives from the assumption that there is a known "reference" topic $t$ that describes completely the information need. For such topic $t$ the retrieved documents can be different compared to the ones observed for a query which represents just one aspect $a$ of the topic.

In Figure 1, we show an example of a number of subtopics found for a topic $t$ and grouped into three clusters, each one with a representative aspect.

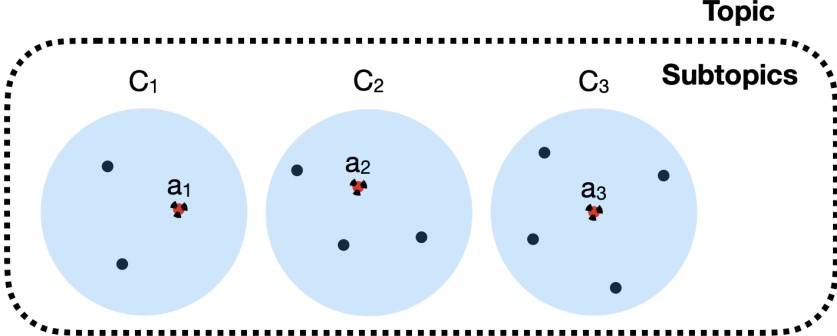

**Figure 1.** An example of clusters of subtopics and aspects.

The Intent-Aware Gain is defined for a set of documents $D$ as:

$$\text{Gain-IA}_t(D) = \sum_{a \in A_t} P(a|t) \cdot \text{Gain}_{t,a}(D) \tag{2}$$

which is a sort of expected value of the gain across the different aspects. $P(a|t)$ is the probability that an aspect $a$ is important to the topic $t$, and $\text{Gain}_{t,a}(D)$ is the gain that can be obtained by the aspect $a$ from the documents D. The importance probability for an aspect of a topic is computed as:

$$P(a|t) = \frac{\text{Imp}_t(a)}{\sum_{a' \in A_t} \text{Imp}_t(a')} \tag{3}$$

while the gain which measures how the documents $D$ retrieved for a query contribute to increment the information relative to a specific aspect of the topic is:

$$\text{Gain}_{t,a}(D) = \left[ 1 - \prod_{d \in D} (1 - \text{Rel}_{t,a}(d)) \right] \tag{4}$$

This last part that is required to compute the Intent-Aware Gain contains the term $\text{Rel}_{t,a}(d)$ which is the relevance degree of a document $d$ with respect to an aspect $a$, estimated as follows:

$$\text{Rel}_{t,a}(d) = \frac{\sum_{s \in C_a} \text{Imp}_t(s) \cdot \text{Rel}_s(d)}{\sum_{s \in C_a} \text{Imp}_t(s)} \tag{5}$$

where $C_a \in C_t$ is the cluster of subtopics belonging to the aspect $a$, and $Rel_s(d)$ is the relevance degree of a document $d$ to a subtopic $s$ estimated as $\text{Rel}_s(d) = 1/\sqrt{\text{Rank}_s(d)}$.

## 3. A Gain for Query Reformulations

Our initial hypothesis in this work is that: (a) we have one information need expressed with different query reformulations, and (b) the topic $t$ is unknown. In particular, given an information need $i$ and its set of reformulations $V_i$, we assume that each reformulation $q \in V_i$ is able to 'reveal' different facets of $i$. Consequently, we need to redefine the expression of the gain of Equation (4) as:

$$\text{Gain}_{i,q}(D) = \left[ 1 - \prod_{d \in D} (1 - \text{Rel}_{i,q}(d)) \right] \tag{6}$$

where $i$ is the *information need* and $q$ is a specific (re)formulation.

The main difference with the original approach, apart from changing variable names, is the fact that (i) we do not have a 'reference' topic $t$ that describes completely the information

need *i*, and (ii) we have one single cluster of query reformulations, or *variants*, $V_i$. For these reasons, we also need an alternative definition of relevance that adapts to our case study:

$$\text{Rel}_{i,q}(d) = \frac{\sum_{s \in V_i} \text{Imp}_q(s)\text{Rel}_s(d)}{\sum_{s \in V_i} \text{Imp}_q(s)} \tag{7}$$

where the relevance of *d*, retrieved by the query variant *q* of the information need *i*, is computed as the weighted average of the relevance of *d* with respect to all the alternative reformulations in $V_i$. The two terms $\text{Imp}_q(s)$ e $\text{Rel}_s(d)$ remain unaltered compared to the previous definitions:

$$\text{Imp}_q(s) = \sum_{d \in D_s^N \cap D_q^N} \frac{1}{\text{Rank}_q(d)} \quad , \quad \text{Rel}_s(d) = \frac{1}{\sqrt{\text{Rank}_s(d)}}$$

### 3.1. A Similarity Matrix for Recall Prediction

In the proposed context, we can think of an 'optimal' query as the one capable of combining all the diverse facets of the information need it represents. In order to estimate which query reformulation *q* is the closest to the unknown optimal one, we propose the following procedure:

1.  we define $D_q$ as the set of documents retrieved by *q*;
2.  $D_i = \bigcup_{q \in V_i} D_q$ as the set of all documents retrieved by *at least* one reformulation *q*;
3.  $\mathbf{R} \in \mathbb{R}^{|V_i| \times |D_i|}$ as the matrix of rankings for the information need *i* where each row corresponds to a specific reformulation and each column to a document. The value of an element $r_{k,d}$ of $\mathbf{R}$ is defined as $|D_q| - \rho_{q,d}$ where $\rho_{q,d}$ is the rank of document *d* retrieved by *q*. $\mathbf{R}$ is at the end normalized with norm *l*2.

At this point, we want to build a similarity matrix to predict the impact in terms of recall that each reformulation will have on the retrieval. We compute the cosine similarity between each pair of rows in $\mathbf{R}$, obtaining a symmetric matrix $\mathbf{S}$ where each row (or column) represents how a reformulation is similar to the others. We use the sum the *k*-th row (or column) of $\mathbf{S}$ to predict the importance of the k-th query; then, we order the query reformulations in decreasing order where greater values indicate a higher probability of retrieving more relevant documents. This measure describes how close each query is to the ideal "centroid" query that perfectly describes the topic.

## 4. Experiments and Analysis

In this section, we describe the analysis of our experiments. In particular, we want to compare the performance in terms of predicted recall among: (i) the gain defined in Equation (6), (ii) an alternative definition that mitigates some arithmetical issues, (iii) and the similarity matrix. To the best of our knowledge, this is the first effort in predicting the recall for the systematic reviews task, when multiple formulations are considered. Therefore, we are not able to directly compare it with an approach explicitly thought for such task. We thus compare our solution with traditional QPP strategies. Furthermore, we use the techniques presented in Umemoto et al. [21] as baselines.

### 4.1. Test Collection and Retrieval Model

The CLEF 2018 eHealth Evaluation Lab Consumer Health Search (CHS) task [20] investigated the problem of retrieving Web pages to support information needs of health consumers that are confronted with a health problem or a medical condition. One subtask (i.e., subtask 3) of this lab is aimed to foster research into building search systems that are robust to query variations (https://github.com/CLEFeHealth/CLEFeHealth2018IRtask, accessed on 15 February 2021).
Queries There are 50 information need for which we have 7 query reformulation for a total of 350 queries: the original 50 queries issued by the general public augmented with 6 query

variations issued individually by 6 research students with no medical knowledge (The queries and the process to obtain them are described in http://www.khresmoi.eu/assets/Deliverables/WP7/KhresmoiD73.pdf, accessed on 15 February 2021).

Collection The collection contains 5,535,120 Web pages and it was created by compiling Web pages of selected domains acquired from the CommonCrawl [20].

Relevance Assessments For each information need, the organizers of the task provided about 500 documents assessed for a total of 25,000 topic-document pairs.

Retrieval Model The index provided by the organizers of the task, an ElasticSearch index version 5.1.1, comes with a standard BM25 model with parameters b = 0.75 and k1 = 1.2 (https://sites.google.com/view/clef-ehealth-2018/task-3-consumer-health-search, accessed on 15 February 2021).

Notice That, among the queries of the CLEF 2018 eHealth CHS collection, the two identified by ids 160006 and 164007 will not retrieve any document in common with the other variants of the same information need (at least for $N \leq 1000$). This is because the text of query 160006 is "nan", while query 164007 has a typo "pros and cons spirculina", instead of spirulina, a type of algae. We stress on this aspect since, for those queries, it will not be possible to compute the value of the gain by definition, since the intersection of their ranked list with the ones for other formulations of the same topic will be empty.

### 4.2. Using Traditional Query Performance Predictors Applied to Recall Prediction for Systematic Reviews

To have a better grasp on the peculiarities of the problem, we first try to apply traditional techniques of Query Performance Prediction (QPP) to our specific setting. We aim at showing that, traditional QPP techniques fail to correctly order formulations when (i) the recall is the key performance indicator; (ii) we sort formulations of the same topic and not queries representing different topics. Showing this, is a further evidence of the importance of using appropriate tools, such as the *gain* as described in Section 2 to correctly tackle the problem. More in detail, we select a set of very well-know QPP models, in order to determine whether they can be satisfactory applied to the prediction of the recall and can be used with the documents and queries that we have at hand. Traditionally, Query Preformance Predictors are divided into two macro-categories, according to the information they exploit to formulate the prediction: Pre-retrieval predictors and Post-retrieval Predictors. *Pre-retrieval predictors* analyze query and corpus statistics prior to retrieval [14,23–28] and *post-retrieval predictors* that also analyze the retrieval results [15,29–36]. Even though Pre-retrieval predictors have the advantage of being faster, since they do not need to retrieve the documents for a certain run, post-retrieval predictors typically perform better. Table 1 reports the predictors that we include in our analyses and a brief description of how they work. It is important to notice that, as for many QPP models, the models that we selected do not actually predict the performance measure. They associate a score to each of the queries, which is expected to correlate with the performance measure, but is on a different scale and cannot be used directly as estimate of the performance.

**Table 1.** Pre- and Post-retrieval predictive baseline models considered.

| Type | Predictor | Description |
|------|-----------|-------------|
| pre-retrieval | max-idf [27] | It considers the maximum value of the idf (inverse document frequency) over the query terms |
| | mean-idf [37] | It computes the mean value of the idf over the query terms |
| | std-idf [37] | It uses the standard deviation of the idf over the query terms |
| | sum-scq [28] | Measures similarity based on cf.idf to the corpus, summed over the query terms. |
| | mean-scq [28] | It relies on the same value of sum-scq, but it normalizes it with the length of the query |
| | max-scq [28] | It relies on the same value of sum-scq, but considers only the maximum value |
| post-retrieval | wig [38] | Standard deviation of the top documents scores in the retrieval list. |
| | nqc [39] | Difference between the mean retrieval score of the top documents, scaled by the score of the entire corpus |
| | smv [40] | It computes the prediction considering the standard deviation of the retrieval scores |

The traditional strategy to evaluate how good a query performance predictor is, consists in computing a traditional retrieval performance measure, such as Average Precision (AP), for each of the query, and determine how much such measure correlates with the prediction scores computed by the QPP model [23–26,28,32–35,38,41–43]. Notice that, there are two main aspects that might impair traditional QPP models in our specific setting:

- Remember that we are in the setting of the systematic reviews. Therefore, it is by far more important to retrieve as many as possible relevant documents, rather than putting them in the first positions. Therefore, we are not interested in estimating the AP, which is a precision based measure, but our aim is to predict which query will have the best recall;
- We do not compare queries meant for different information needs, which is the typical evaluation scenario for QPP models.

On the other hand, we aim at understanding which one, among a set of queries representing the same information need, achieve the best result.

To determine whether we are impaired by the first problem, we first apply the traditional QPP considering only the default formulation of each topic, and we compare whether the predictors are capable of correctly determining the inter-topic performance. More in detail, with this first experiment, we are interested in understanding whether the baseline predictors are capable of predicting which *topic* will have the best recall, using a single formulation for each of them. Table 2 reports the result of such analysis.

We can observe that, by looking at Table 2, the results are in line with previous similar experiments in the literature, such as [5,44]. Almost all the predictors are able to achieve a significant correlation with the recall (with level $\alpha = 0.01$). Two noticeable exceptions are represented by nqc and smv: traditionally, they are considered among the best predictors, but in this specific scenario they fail, with correlations not statistically different from 0. Our hypothesis is that, while pre-retrieval predictors tend to be estimators of the recall base of a query, and therefore tend to correlate with the recall itself, post-retrieval predictors tend to compute their predictors based on the scores that the retrieval model assigns to the top-ranked documents. In this sense, post-retrieval predictors are "top-heavy": they focus on the upper part of the ranked list of documents. This behaviour favours predicting the performance for top-heavy measures, such as Average Precision or nDCG. Instead, our task consists in predicting the recall, given a *long* list of documents. It is not unlikely that the

upper part of the list of retrieved documents is saturated with relevant ones; nevertheless, we are more interested in being sure that *every* relevant document has been considered, rather than saying whether the top part of the ranked list contains relevant documents.

**Table 2.** Kendall's $\tau$ correlation observed between recall and prediction scores for both pre- and post-retrieval traditional predictors, if we compare the default formulations of different topics. Results are in line with correlation values previously observed in other scenarios. The symbol † indicates that the correlation is statistically greater than 0 at level $\alpha = 0.05$, while the ‡ indicates a significance level of 0.01, the absence of any symbol indicates that results cannot be deemed statistically greater than 0. We compute the Kendall's $\tau$ correlation at different cutoff levels of the ranked lists (100, 1000, and 10,000).

| Type | Predictor | Kendall's $\tau$ | | |
|---|---|---|---|---|
| | | **100** | **1000** | **10,000** |
| | max-idf | 0.3185 ‡ | 0.3260 ‡ | 0.2875 ‡ |
| | mean-idf | 0.2996 ‡ | 0.3218 ‡ | 0.2555 ‡ |
| | std-idf | 0.2947 ‡ | 0.2989 ‡ | 0.2343 † |
| pre-retrieval | sum-scq | 0.2637 ‡ | 0.2581 ‡ | 0.1739 |
| | mean-scq | 0.3479 ‡ | 0.3652 ‡ | 0.3299 ‡ |
| | max-scq | 0.3502 ‡ | 0.3724 ‡ | 0.2833 ‡ |
| | wig | 0.3029 ‡ | 0.3218 ‡ | 0.2882 ‡ |
| post-retrieval | nqc | 0.2865 ‡ | 0.1911 | 0.1135 |
| | smv | 0.1797 | 0.1332 | 0.0229 |

We now switch the focus from predicting the performance *across* topics, to predict the performance *within* topics. Instead of comparing the performance that the standard formulation is expected to achieve for each topic, we try to sort different formulations for the same topic, according to the predicted performance. Table 3 reports the results of our analysis.

Compared to the results observed in Table 2, the performance achieved by traditional predictors for the "within"-topics prediction, is extremely lower, with very few cases of significantly positive correlation between the predicted and observed recall. Note that, even though we agree with [13] on the fact that predicting the best query among a series of formulations of a topic is a hard task, we end up with diametrically opposite conclusions. Scells et al. [13] observed severe flaws in traditional QPP techniques when *predicting the performance across topics*. On the other hand, they found the task of predicting the performance within topics (which they refer to as Query Variation Performance Prediction (QVPP)) to be easier, achieving higher (although still very low) results. What we observe here, is diametrically opposite: we found the worst results when predicting results within topics, and performance in line with previous literature for the predictions across topics. A possible explanation for this phenomenon is that we use the traditional QPP models for a different task compared to Scells et al. [13]. In fact, our aim is to predict the recall, while Scells et al. [13] aim at predicting the Average Precision. As a final remark, we want to point out that, Zendel et al. [45] recently showed how the "QVPP" is a harder task, compared to traditional QPP, confirming in this sense our findings.

### 4.3. Analysis of the Results

Given what we observed in Section 4.2, we are interested in understanding whether the GAIN-based proposed by [21] (cfr. Equation (4)) can overcome the problems in this specific setting shown by traditional QPP models. The results are shown in Figure 2a,d,h. Each figure is divided into two parts: top, we show the distribution of values of the GAIN (or similarity), ordered increasingly, for each query reformulation (350 in total); bottom, we plot for each topic (50 topics) the value of the correlation Kendall $\tau$ between the query reformulations ordered by decreasing GAIN (or similarity) and the reformulations ordered

by decreasing true recall. The blue dots indicate a statistically significant correlation greater (or lower) than zero, while black dots the topics for which it is not possible to compute the correlation.

**Table 3.** Performance achieved by traditional predictors, applied to our specific case. Each predictor has been used to predict the performance of the different formulations. We report the mean score and standard deviation of the correlation computed over the different topics. We also report the first quartile, third quartile and number of topics (over the 50 available) for which the correlation between the predicted and observed recalls for their (re)formulations is significantly greater than 0.

| Type | Predictor | Cutoff | Q1 | Kendall's $\tau$ Mean (Std) | Q3 | Sign. |
|------|-----------|--------|-----|----------|-----|-------|
| pre-retrieval | max-idf | 100 | −0.5417 | −0.1085 (0.4825) | 0.1183 | 3 |
| | | 1000 | −0.5295 | −0.0973 (0.4755) | 0.2263 | 1 |
| | | 10,000 | −0.5699 | −0.1227 (0.4628) | 0.1584 | 2 |
| | mean-idf | 100 | −0.4214 | −0.0449 (0.5272) | 0.3333 | 4 |
| | | 1000 | −0.4821 | −0.0617 (0.4898) | 0.2167 | 4 |
| | | 10,000 | −0.4214 | −0.0549 (0.4698) | 0.2473 | 3 |
| | std-idf | 100 | −0.4880 | −0.1606 (0.4927) | 0.0915 | 4 |
| | | 1000 | −0.4190 | −0.0999 (0.5107) | 0.1938 | 5 |
| | | 10,000 | −0.4064 | −0.1537 (0.4347) | 0.1576 | 1 |
| | sum-scq | 100 | −0.2381 | −0.0102 (0.4021) | 0.2381 | 1 |
| | | 1000 | −0.2985 | 0.0893 (0.4276) | 0.4000 | 4 |
| | | 10,000 | −0.3126 | 0.0150 (0.4558) | 0.2750 | 5 |
| | mean-scq | 100 | −0.3250 | 0.0322 (0.5231) | 0.3901 | 6 |
| | | 1000 | −0.3898 | 0.0135 (0.4838) | 0.3333 | 5 |
| | | 10,000 | −0.3250 | 0.0005 (0.4505) | 0.2985 | 3 |
| | max-scq | 100 | −0.3541 | −0.0369 (0.4333) | 0.2765 | 1 |
| | | 1000 | −0.3341 | −0.0312 (0.4447) | 0.2568 | 2 |
| | | 10,000 | −0.3459 | −0.0484 (0.4441) | 0.1912 | 2 |
| post-retrieval | wig | 100 | −0.6790 | −0.1743 (0.5206) | 0.1376 | 4 |
| | | 1000 | −0.4088 | −0.0266 (0.5031) | 0.2519 | 6 |
| | | 10,000 | −0.4214 | 0.0171 (0.5185) | 0.3849 | 6 |
| | nqc | 100 | −0.5611 | −0.0880 (0.5554) | 0.2381 | 6 |
| | | 1000 | −0.4405 | −0.1244 (0.5004) | 0.1511 | 4 |
| | | 10,000 | −0.4850 | −0.1539 (0.4991) | 0.1539 | 3 |
| | smv | 100 | −0.5621 | −0.1653 (0.4836) | 0.1849 | 1 |
| | | 1000 | −0.5542 | −0.1626 (0.4604) | 0.1859 | 0 |
| | | 10,000 | −0.6243 | −0.2207 (0.4882) | 0.0994 | 2 |

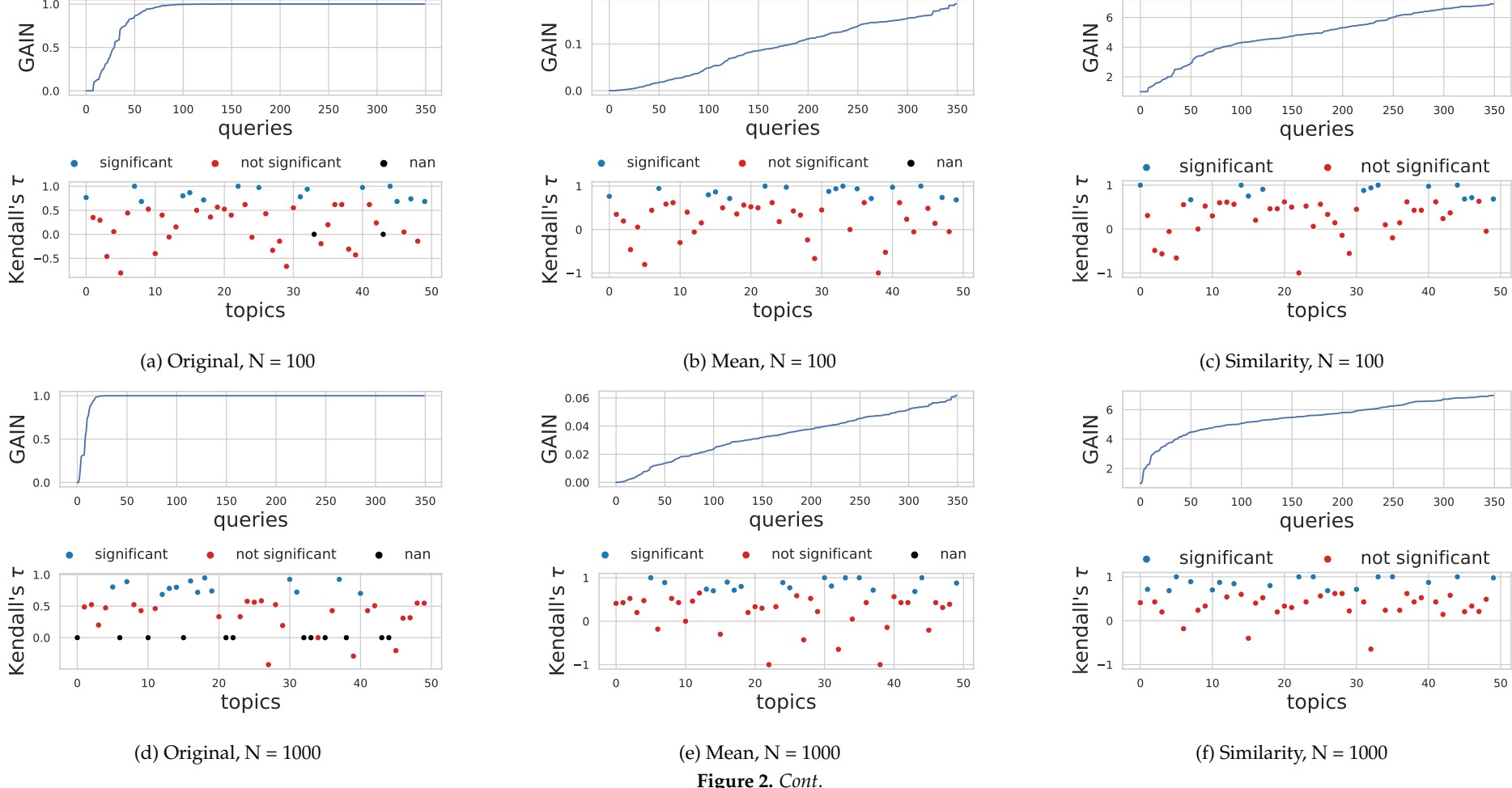

(a) Original, N = 100

(b) Mean, N = 100

(c) Similarity, N = 100

(d) Original, N = 1000

(e) Mean, N = 1000

(f) Similarity, N = 1000

**Figure 2.** *Cont.*

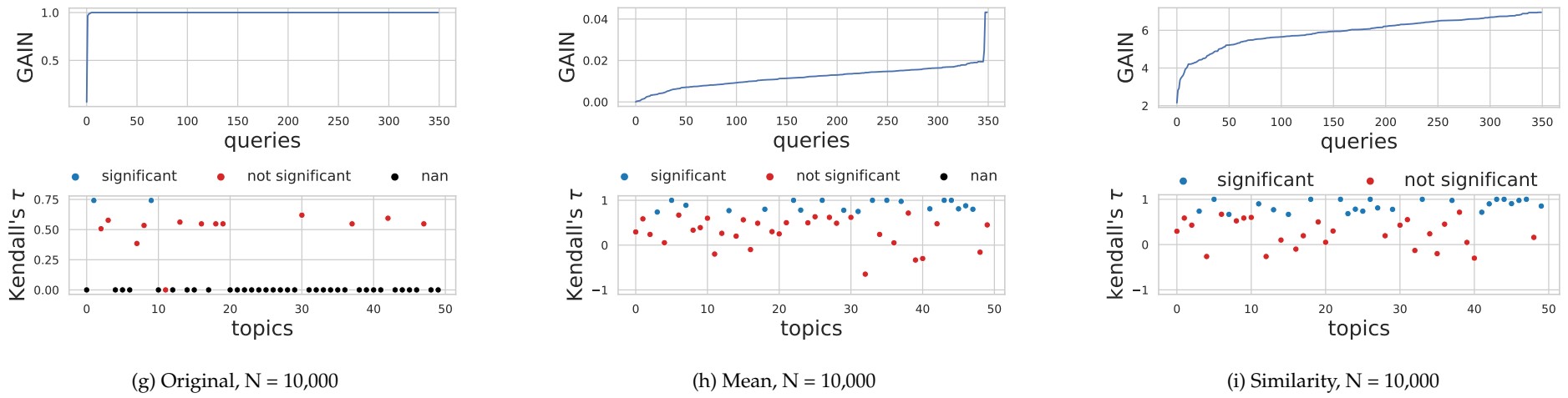

(g) Original, N = 10,000                (h) Mean, N = 10,000                (i) Similarity, N = 10,000

**Figure 2.** The three rows show the results for the measures computed at N = 100, 1000, and 10,000, respectively. The three columns represent results of the GAIN as proposed in [21], the mean aggregation and the similarity-based aggregation strategy. Each subfigure shows (top) the distribution of the GAIN, ordered increasingly, of the 350 queries and (bottom) correlation between the reformulations ordered by predicted GAIN (or similarity) and the reformulations ordered by the true recall.

### 4.3.1. Saturated GAIN Distribution

In Figure 2a,d,h, we show that the value of the gain saturates to 1 for most query reformulation. This is more evident when we increase the number of documents $N$ of Equation (6) from $N = 100$ up to $N = 10,000$. This behavior, due to the importance in Equation (1) that multiplies $N$ numbers less than one, makes the GAIN not useful to discriminate the different query variants of an information need, since every variant will have gain equal to 1. In addition, when all the reformulations have the same gain, it is impossible to compute the Kendall $\tau$ correlation to predict the performance (black dots with correlation value 0 in the figure). Being not saturated is not by itself a desirable feature for the gain measure. Nevertheless, the faster the gain saturates, the harder it is to discriminate between different formulations. In this sense, a GAIN measure capable of spreading better the options in the entire domain is preferable.

### 4.3.2. Alternative GAIN Definition

In order to mitigate the aforementioned problems, we propose an alternative definition of the gain of Equation (6) substituting the product with an average:

$$GAIN_{i,q}(D) = \left[1 - \frac{\sum_{d \in D}(1 - \text{Rel}_{i,q}(d))}{|D|}\right] \tag{8}$$

The results of this new formulation are shown in Figure 2b,e,g. The distribution of the gain is more spread across all the reformulations and does not saturate to one. There is also a more stable prediction of the performances for each topic: the number of statistically significant predictions of the recall of the reformulation is between 17 and 19, from $N = 100$ and $N = 10,000$; in addition, the number of negative correlations (wrong predictions of performance) decreases. This indicates (as we may expect) that with more information (more documents, greater $N$) we can predict better the order of importance, in terms of recall, of each reformulation.

### 4.3.3. Using Similarity Matrix for Recall Prediction

In Figure 2c,f,i, we show the ability to predict the performance of a query reformulation using the correlation between the similarity-based approach presented in Section 3.1. The values of the Similarity are spread and do not saturate to the maximum value of the sum of a row of $S$ (in our experiments equal to 7). By increasing the number $N$ of documents, we improve the capability to predict the performance of the query reformulation; in particular, there are no statistically significant negative correlation and the total number of negative correlations decreases from $N = 100$ to $N = 10,000$.

Besides the qualitative aspects, Table 4 reports also the numerical performance comparison between the GAIN as proposed by [21], its version which employs the mean, and the similarity-based gain.

### 4.3.4. Final Remarks

In this last section of the analysis of the results, we want to briefly summarize our findings. As a remainder, we want to point out that, the GAIN measure proposed by [21], was originally used to estimate the missing information that the user could have gained, by using different subtopic formulations, showed in a user-interface. Although such task shares similar aspects with the one of predicting the recall, they are not fully overlapping. Our main contributions in this paper are:

- First, adapting an already established technique to a different task. In this sense, to the best of our knowledge, this is the first effort in adapting the GAIN measure proposed by Umemoto et al. [21] to the query formulation recall prediction task.
- Secondly, its "mean" version, which we refer to as "Mean Gain", is observed here for the first time, as a better adaptation of [21] to the predictive task.

- Finally, the Similarity-based Gain is a completely new contribution of this manuscript, which exploits similar elements to the gain measure proposed by Umemoto et al. [21].

**Table 4.** Kendall's $\tau$ correlation observed for the task of predicting the query formulation recall, using the similarity based approaches. In bolt, best mean score for each cutoff. Note that all the methods considered perform better than traditional predictors(cft Table 3). We also have a higher number of significant rankings compared to the one observed before.

| | | | Kendall's $\tau$ | | |
|---|---|---|---|---|---|
| **Predictor** | **Cutoff** | **Q1** | **Mean (Std)** | **Q3** | **Sign.** |
| Original Gain [21] | 100 | 0.0000 | 0.3422 (0.4696) | 0.6831 | 15 |
| | 1000 | 0.0000 | 0.3783 (0.3527) | 0.6609 | 13 |
| | 10,000 | 0.0000 | 0.1600 (0.2606) | 0.4765 | 2 |
| Mean Gain | 100 | 0.1456 | 0.3822 (0.4936) | 0.7320 | 16 |
| | 1000 | 0.2417 | 0.4042 (0.4796) | 0.7320 | 17 |
| | 10,000 | 0.2709 | 0.5111 (0.3984) | 0.8000 | 19 |
| Similarity based Gain | 100 | 0.1429 | 0.3768 (0.4636) | 0.6581 | 13 |
| | 1000 | 0.3083 | 0.5069 (0.3505) | 0.7143 | 17 |
| | 10,000 | 0.2521 | 0.5443 (0.3930) | 0.8876 | 24 |

Table 4 shows that the similarity based gain has the overall best performance both compared to other gain based measures and traditional predictors (cfr. Table 3). Interestingly, while the original gain worsen with the increase of the cutoff (as observed both in Tables 2 and 3), both the mean based and the similarity one tend to improve their performance when the cutoff increase. The original gain suffers of the "saturated gain", as reported in Section 4.3.1, while our proposal (both mean and similarity) improve as new relevant information is added.

## 5. Conclusions and Future Work

In this paper, we have presented a study that evaluates different definitions of the GAIN of a reformulation for an information need. We adapted the definition of gain proposed by Umemoto et al. [21] to the context of Consumer Health Search, and we used a standard test collection to evaluate our hypotheses: can we use the gain metric to predict the performance of each reformulation? Is there a better formulation that can produce an order of the importance of each reformulation in terms of recall?

We found that for recall based tasks where the number of documents to retrieve may be large, $N > 100$, the original definition of GAIN saturates quickly to 1. We proposed an alternative definition that mitigates this problem, and we also presented a similarity based approach that tries to capture the 'optimal' query reformulation among all the available formulations of an information need. The analysis of the results confirms that our approach significantly improves the prediction of the order of importance of each reformulation in terms of recall.

In conclusion, the proposed technique is meant to help practitioners in tackling the systematic reviewing task. In this sense, our technique is not meant for a general-purpose query suggestion strategy (more in line with the model proposed by Umemoto et al. [21]). The described approach is meant to boost queries with higher recall, in a context where the practitioner is, in a sense, forced to explore a large number of documents. Assuming that a practitioner needs to review *all* the documents about a specific topic, it is vital to reduce as possible time spent reviewing documents. Therefore, our technique can help in determining, among a series of queries for the very same information need, which one is more likely to return the most relevant documents. Being able to explore first more promising queries, can greatly speed up the systematic reviewing process. Among our future work, we plan to further investigate the robustness and generalizability of the approach. In particular, we plan to include new collections, topics and formulations. We plan

to investigate the news domain through multiple formulations available in the UQV100 collection [1]. We are currently investigating the possibility to smooth the contribution of each reformulation in the similarity matrix $S$ with a *locality* parameter $w$. This parameter can be used as an exponent for each element of $S$ and decide whether to get reformulations closer, $w < 1$, or push them far away, $w > 1$, to create sub-clusters of reformulations and obtain a better prediction.

**Author Contributions:** Conceptualization, G.M.D.N. and G.F.; methodology, G.M.D.N. and G.F.; software, G.M.D.N. and G.F.; validation, G.M.D.N. and G.F.; formal analysis, G.M.D.N. and G.F.; investigation, G.M.D.N. and G.F.; resources, G.M.D.N. and G.F.; data curation, G.M.D.N. and G.F.; writing—original draft preparation, G.M.D.N. and G.F.; writing—review and editing, G.M.D.N. and G.F.; visualization, G.M.D.N. and G.F.; supervision, G.M.D.N. All authors have read and agreed to the published version of the manuscript.

**Funding:** This research received no external funding.

**Institutional Review Board Statement:** Not applicable.

**Informed Consent Statement:** Not applicable.

**Data Availability Statement:** The code used is publicly available at: https://github.com/guglielmof/A-Study-of-a-Gain-Based-Approach the data can be found at: https://github.com/CLEFeHealth/CLEFeHealth2018IRtask and the index is available at: https://sites.google.com/view/clef-ehealth-2018/task-3-consumer-health-search accessed on 8 September 2021.

**Conflicts of Interest:** The authors declare no conflict of interest.

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
