# Peer review of "A Study of a Gain Based Approach for Query Aspects in Recall Oriented Tasks"

_applsci, doi:10.3390/app11199075_

Round 1

Reviewer 1 Report

The topic undertaken is interesting, up-to-date and meaningful, as it addresses the problem “of building search engines that are robust to query variations to support information needs of health consumers”. The study is well rooted in the current literature and its contribution to this literature is convincingly  presented. There is a novel tool proposed (or rather an adaptation and slight modification of an existing intent-aware metric) and there is an experimental analysis carried on to present its good performance. However, there are some issues that could be addressed in the revision.

The manuscript is difficult to follow and there are many ambiguities.  Terms and context obvious for the Authors may not be evident for the readers, so I would suggest including more explanations. For example, the content of  “Caveat” on page 5 is not clear for the outsiders. Simpler language could be helpful, avoiding complex language structure like “ they show that they show how they” on page 1. Or, what does “id f” in Table 1 stand for?

I lack the robustness check – the proposed metrics are demonstrated in a specific environment, are they going to perform similarly well with different set of information needs/topics, queries and documents? Additionally, the Authors report different than in literature findings regarding the use of standard QPP for their data (worse performance within topic than across topics) – what is a possible reason for such differences?

In the conclusion of the manuscript, I would suggest a discussion regarding specific possibilities of practical implementation of the research findings. For example, I would look for the clarification how using the proposed metric will help to increase gain in terms of importance, relevance and novelty regarding the practical aim to fulfill the information needs of health consumers. How could the proposed metrics improve the functioning of the search systems based on automatic queries reformulations?

Author Response

We thank the reviewer for the fruitful comments. 

We give a detailed reply here.

R1.1 The manuscript is difficult to follow and there are many ambiguities.  Terms and context obvious for the Authors may not be evident for the readers, so I would suggest including more explanations. For example, the content of  “Caveat” on page 5 is not clear for the outsiders. Simpler language could be helpful, avoiding complex language structure like “ they show that they show how they” on page 1. Or, what does “id f” in Table 1 stand for?

(A) We thank you the reviewer for this suggestion. We have removed the unnecessary complex terms and expanded the acronyms. We also reviewed all the text for English mistakes (such as “they show that they show …).

R1.2 I lack the robustness check – the proposed metrics are demonstrated in a specific environment, are they going to perform similarly well with different set of information needs/topics, queries and documents?

(A) We agree with the reviewer, and we carefully thought about this issue. Due to the very short time for the review of the paper, it was not possible to set up a new experimental analysis with a different experimental collection. We also thought about randomly sampling a subset of query reformulations to simulate different situations, but this would contrast with the difficulty of the task of ranking the variants (the less variants the easier to correlate with the optimal rank). For these reasons, we added in the last section the description of a larger set of longitudinal experiments that will evaluate the robustness of all the QPP measures as well as QVPP.

R1.3 Additionally, the Authors report different than in literature findings regarding the use of standard QPP for their data (worse performance within topic than across topics) – what is a possible reason for such differences?

(A) We were surprised too by this large difference in the scores of the QPP and QVPP. At the beginning, we double checked that the source code was not bugged. This is the same source code used in [Guglielmo Faggioli, et al.: An Enhanced Evaluation Framework for Query Performance Prediction. ECIR (1) 2021: 115-129] and our results are in-line with our previous findings. Apossible explanation for this phenomenon is that we use the traditional QPP models for a different task compared to Scells et al.[14]. In fact, our aim is to predict the recall, while Scellset al.[14] aim at predicting the Average recision. We added this comment in the paper.

R1.4 In the conclusion of the manuscript, I would suggest a discussion regarding specific possibilities of practical implementation of the research findings. For example, I would look for the clarification how using the proposed metric will help to increase gain in terms of importance, relevance and novelty regarding the practical aim to fulfill the information needs of health consumers. How could the proposed metrics improve the functioning of the search systems based on automatic queries reformulations?

(A) We thank you the reviewer for this suggestion. We added a paragraph in the conclusions to discuss these matters.

Reviewer 2 Report

This paper analyzes the issue of producing Consumer Health Search (CHS) system based on query variations to gain high recall and related information. They study also an alternative formulation of the intent-aware metric proposed by Umemoto et al.

They adapted the GAIN measure proposed by Umemoto et al. to the query formulation recall prediction task. Their main contribution was introducing of the similarity-based Gain. Experimental evaluations were performed well and comprehensively.

Why did you apply traditional QPP models to your specific settings?

Are there related approaches to yours? I would expect to provide some related approaches?  

Here are some minor revisions:

  1. In line 5: “an effective Consumer Health Search (CHS) systems” should be corrected as “an effective Consumer Health Search (CHS) system” or “effective Consumer Health Search (CHS) systems”.
  2. In line 32: “they show that they show” should be corrected, “they show” is enough.
  3. In line 83: “is a the cluster of subtopics” sentence should be corrected.
  4. In line 86: “q a specific (re)formulation” should be “q is a specific (re)formulation”?
  5. In line 161: Why you don’t need to compute AP? Why you just considered the recall why you didn’t consider the precision as well?
  6. In line 172: there must be full stop at the end of the sentence.
  7. In Table 2: what is “100, 1000, 10000? They are not defined before using (I notice they are number of document but it should be clearly defined for reader before using).
  8. In Table 2: “The symbol † indicates that the correlation is statistically greater than 0 at level α = 0.05, while the ‡ indicates a significance level of 0.01”. What does it mean if there is not any symbol?
  9. “References” word is missing in References section

Author Response

Thank you very much for the constructive comments.

Please, find hereby a detailed set of answers.

R2.0 Why did you apply traditional QPP models to your specific settings? Are there related approaches to yours? I would expect to provide some related approaches?

(A) We applied QPP and QVPP as a baseline since these are the approaches traditionally used to perform query performance prediction. The difference with our task is that, in general Q(V)PP are used to predict measures that are more precision oriented, while our task is recall oriented. Therefore, these measures are the approaches with which we need to compare with. The innovative aspect of our work is to transform an information-gain measure into a new query performance prediction measure.

R2.1 In line 5: “an effective Consumer Health Search (CHS) systems” should be corrected as “an effective Consumer Health Search (CHS) system” or “effective Consumer Health Search (CHS) systems”.

(A) Fixed. Thank you.

R2.2 In line 32: “they show that they show” should be corrected, “they show” is enough.

(A) Fixed. Thank you.

R2.3 In line 83: “is a the cluster of subtopics” sentence should be corrected.

(A) Fixed. Thank you.

R2.4 In line 86: “q a specific (re)formulation” should be “q is a specific (re)formulation”?

(A) Fixed. Thank you.

R2.5 In line 161: Why you don’t need to compute AP? Why you just considered the recall why you didn’t consider the precision as well?

(A) Since we are approaching a high recall task, we do not aim at precision. We have added a paragraph to explain and clarify this aspect.

R2.6 In line 172: there must be full stop at the end of the sentence.

(A) Fixed it. Thank you.

R2.7 In Table 2: what is “100, 1000, 10000They are not defined before using (I notice they are number of document but it should be clearly defined for reader before using).

(A) Thank you for this suggestion. We added the explanation of this value in the caption of the table.

R2.8 In Table 2: “The symbol † indicates that the correlation is statistically greater than 0 at level α = 0.05, while the ‡ indicates a significance level of 0.01”. What does it mean if there is not any symbol?

(A) We fixed this missing information and added an explanation in the caption.

R2.9 “References” word is missing in References section

(A) Fixed it. Thank you.